# Dietetic-Led Nutrition Interventions in Patients with COVID-19 during Intensive Care and Ward-Based Rehabilitation: A Single-Center Observational Study

**DOI:** 10.3390/nu14051062

**Published:** 2022-03-03

**Authors:** Ella Terblanche, Jessica Hills, Edie Russell, Rhiannon Lewis, Louise Rose

**Affiliations:** 1Dietetics Department, St Georges University Hospitals NHS Foundation Trust, London SE1 8WA, UK; jessica.hills@uon.edu.au (J.H.); edie.russell@stgeorges.nhs.uk (E.R.); r.lewis3@rbht.nhs.uk (R.L.); 2Florence Nightingale Faculty of Nursing, Midwifery and Palliative Care, King’s College London, London WC2R 2LS, UK; louise.rose@kcl.ac.uk; 3Adult Critical Care, Guy’s and St Thomas’ NHS Foundation Trust, London SE1 7EH, UK

**Keywords:** COVID-19, intensive care, dietitian/dietician, nutrition, malnutrition, weight loss

## Abstract

Background: In this study, a report of dietitian-led nutrition interventions for patients with COVID-19 during ICU and ward-based rehabilitation is provided. As knowledge of COVID-19 and its medical treatments evolved through the course of the pandemic, dietetic-led interventions were compared between surge 1 (S1) and surge 2 (S2). Methods: A prospective observational study was conducted of patients admitted to the ICU service in a large academic hospital (London, UK). Clinical and nutrition data were collected during the first surge (March–June 2020; *n* = 200) and the second surge (November 2020–March 2021; *n* = 253) of COVID-19. Results: A total of 453 patients were recruited. All required individualized dietetic-led interventions during ICU admission as the ICU nutrition protocol did not meet nutritional needs. Feed adjustments for deranged renal function (*p* = 0.001) and propofol calories (*p* = 0.001) were more common in S1, whereas adjustment for gastrointestinal dysfunction was more common in S2 (*p* = 0.001). One-third of all patients were malnourished on ICU admission, and all lost weight in ICU, with a mean (SD) total percentage loss of 8.8% (6.9%). Further weight loss was prevented over the remaining hospital stay with continued dietetic-led interventions. Conclusions: COVID-19 patients have complex nutritional needs due to malnutrition on admission and ongoing weight loss. Disease complexity and evolving nature of medical management required multifaceted dietetic-led nutritional strategies, which differed between surges.

## 1. Introduction

In March 2020, the exponential increase in intensive care unit (ICU) admissions in the United Kingdom (UK) due to the COVID-19 pandemic required significant planning and restructuring of dietetic services to ensure safe and effective nutrition provision [1]. ICU dietitians were faced with multiple challenges including how best to provide nutrition support for patients with an unknown disease, rapidly train redeployed dietitians inexperienced in ICU nutrition, prioritize dietetic-led nutrition interventions, and manage the logistics of shortages of enteral feed, feed pumps, and ancillaries. Critically ill patients with COVID-19 appeared nutritionally complex, with no international consensus on optimal nutritional management.

COVID-19 patients frequently present with malnutrition [2,3]. This is due to pre-existing chronic disease associated with underlying poor nutritional intake, combined with a further decline due to common COVID-19 symptoms including gastrointestinal dysfunction and loss of taste and smell [2]. Patients with severe COVID-19 pneumonia exhibit a marked hyperdynamic state with persistent pyrexia leading to hypermetabolism and protein catabolism [4,5]. Furthermore, enteral nutrition (EN) intolerance was reported due to gastrointestinal symptoms, refractory hypoxemia requiring prone positioning, hypotension or shock requiring the use of vasopressors, and the progression of multiple organ failure [1,6,7,8].

Nutritional guidelines were written rapidly based on experiential learning and knowledge gained from dietitians in other countries managing critically ill patients with COVID-19, in addition to prior dietetic knowledge of managing patients with severe respiratory failure [1,6,7,8]. Feeding protocols were devised to simplify nutrition delivery and ensure consistency of nutrition interventions [6,7,8,9] at a time when EN was perceived difficult to achieve and not a medical priority [9].

The aim of this study was to describe dietitian-led nutrition interventions for patients with COVID-19 during ICU admission and ward-based rehabilitation in a large tertiary ICU service in a single center. As our knowledge of COVID-19 and its medical treatments was evolving, we compared dietetic-led nutrition interventions between surge one (S1) and surge two (S2) in the UK.

## 2. Materials and Methods

### 2.1. Study Design, Setting, and Sample

A prospective, observational study was conducted by enrolling critically ill patients with COVID-19 admitted to an ICU (66 beds outside of pandemic conditions, 148 maximum bed number during pandemic) in a large academic hospital in London, UK. Data were collected during S1 (March–June 2020) and S2 (November 2020–March 2021).

All adult (≥16 years) patients with COVID-19 who received advanced respiratory support defined as invasive ventilation via endotracheal or tracheostomy tube and required EN or parenteral nutrition (PN) for longer than 48 h in ICU were included. Data were collected from ICU admission to hospital discharge.

### 2.2. Nutritional Interventions

To support increased ICU patient numbers, dietitians were redeployed from other clinical areas, equating to one dietitian to 25 ICU patients. The primary treatment aim was to commence nutrition using the newly devised local hospital COVID-19 ICU feeding protocol with EN within 48 h of admission. The protocol advised feeding to start with a high protein EN product, at 30 mL/h for six hours, after which time a gastric residual volume (GRV) should be obtained. If the volume was less than 500 mL (or 300 mL if in the prone position), the rate of feeding was increased to meet a target rate based on the patient’s weight, (actual or ideal if BMI > 25 kg/m^2^). Ideal body weight (IBW) was based on the patient’s height calculated to BMI of 25 kg/m^2^ [10]. If EN was not tolerated sufficiently, as defined in the protocol, PN was commenced. As per usual practice prior to COVID-19, all patients were screened by a dietitian each morning, and only patients whose nutritional needs were not met using the ICU feeding protocol underwent dietetic assessment within 48 h of commencing nutrition. Treating dietitians set energy and protein targets as per the European Society for Parenteral and Enteral Nutrition (ESPEN) guidelines [7,10], with targeted energy recommended at 20 kcal/kg/day to avoid overfeeding in the early phase and increased to 30 kcal/kg/day or more to facilitate rehabilitation after the acute phase of critical illness. Actual weight was used for patients with a BMI less than 25 kg/m^2^, IBW for those with a BMI of 25–30 kg/m^2^, and adjusted body weight for those with BMI more than 30 kg /m^2^. Indirect calorimetry was used where appropriate to aid the prediction of energy needs during the recovery phase in ICU. Nutritional data were collected on type of EN, use of protein supplementation, reasons for dietetic-led nutrition interventions during the ICU admission, on ICU discharge, during ward admission, and at hospital discharge.

### 2.3. Data Collection

Baseline demographics were collected from the medical record including pre-existing comorbidities, the severity of illness using the Acute Physiology and Chronic Health Evaluation (APACHE) II score, anthropometric measurements, and malnutrition risk as defined by Global Leadership Initiative on Malnutrition [11] and European ICU criteria [10]. Outcome data included ICU mortality and length of stay in ICU and hospital.

Anthropometric measurements included body mass index (BMI) calculated using ICU admission body weight (kg) and height (m). When a recent accurate weight was not available from medical records, we measured patient weight using a hoist or patient transfer scales or obtained a reported weight from a family member. When height was not available, it was estimated from ulnar length measurement and converted into an estimated height [12]. ICU survivors were weighed at ICU and hospital discharge using sitting, standing, or hoist scales. Change in weight during ICU and hospital stay was determined in kilograms and percentage total weight loss. Percentage weight loss ≥5% was considered clinically significant and was used to diagnose malnutrition [11].

### 2.4. Data Analysis

Categorical data are presented as frequencies and proportions and continuous data as means and standard deviations (SD). Proportions were compared using chi-squared or Fisher’s exact test (depending on cell size) and continuous data using a two-sample independent *t*-test or the Wilcoxon rank-sum (Mann–Whitney) test. All tests were two-tailed, with *p* ≤ 0.05 considered statistically significant. Statistical analyses were performed using Stata version 17 (StataCorp LLC, College Station, TX, USA).The STROBE reporting guidelines for observational studies were followed.

### 2.5. Ethical Considerations

The study was approved as a service evaluation by the National Health Service Health Research Authority and the Research and Development Service at St George’s University Hospitals NHS Foundation Trust, London, UK (Registration Number AUDI000637). Consent was waived, as the project was approved as a service evaluation.

## 3. Results

There were 453 critically ill patients with COVID-19, of which 200 patients were admitted during S1 (March–June 2020) and 253 during S2 (November 2020–March 2021). Baseline characteristics are shown in Table 1. Figure 1 presents patient numbers from admission to discharge.

### 3.1. Patient Characteristics

S1 patients were younger than in S2, with a mean (SD) of 57.9 (12.7) years, compared with 62.5 (12.0) years (*p* = 0.001), and had fewer comorbidities (*p* = 0.02). Length of ICU stay was longer in S2 (*p* = 0.019) (Table 1). Of the 453 patients, 167 (37%) were malnourished on ICU admission, with 64 (32%) in S1 and 103 (40%) in S2.

### 3.2. Dietitian-Led Nutrition Interventions

Individualized dietetic-led nutrition interventions were required by all 453 (100%) patients during ICU admission, as nutritional needs were not met using the standardized ICU nutrition protocol. Mean (SD) time to first dietetic-led nutrition interventions was 2.3 (1.2) days in S1 and 2.9 (2.3) days in S2. Patients received a mean (SD) of 5.2 (4.5) dietetic interventions during the ICU stay. Patients had a similar number of dietetic-led nutrition interventions during their ICU stay in S1 and S2 (mean 5.2 (4.2) compared with 5.3 (4.8) times). PN was only required for six (1%) patients during S2 and none in S1.

More patients required feed adjustment for calories derived from propofol, impaired renal function, or changes in fluid or electrolyte status in S1, whereas more patients needed adjustments for gastrointestinal dysfunction in S2 (Table 2).

Most patients required high-protein enteral feeds during ICU admission, with similar proportions in S1 and S2 (184, 92% vs. 236, 93%). There was no difference in the proportion of patients requiring concentrated feeds (39, 20% in S1 vs. 45, 18% in S2). More peptide feeds were used in S1 (26, 13%) than in S2 (17, 7%) (*p* = 0.03). Protein supplementation also increased from S1 (98, 49%) to S2 (174, 69%) (*p* = 0.001). During S1, the first choice of feed was not available, due to supplier production shortages in 61 (30%) patients, whereas shortages were not experienced during S2.

### 3.3. Weight Loss

Mean (SD) weight loss over the ICU admission was 7.9 kg (6.8 kg), equivalent to a mean (SD) total percentage loss of 8.8% (6.9%), indicating clinically significant malnutrition. Mean (SD) weight loss over the total hospital stay was 7.5 kg (6.6 kg), suggesting no further weight loss occurred after ICU discharge. Details of weight loss according to surge are shown in Table 3.

### 3.4. Dietetic-Led Nutrition Interventions after ICU Discharge

Ward dietitians received a dietetic handover for all patients discharged alive from ICU (*n* = 177). Most patients (160, 90%) received ward-based dietetic-led nutrition interventions, with similar numbers in S1 (78, 89%) and S2 (82, 92%). The 17 patients not reviewed were discharged home before the ward dietitians were able to assess.

Upon discharge to the ward, patients were reviewed earlier during S1 than during S2, within a mean (SD) of 1.9 (1.3) vs. 2.4 (1.8) days (*p* = 0.04), but at similar frequencies (2.8 (2.5) times over 9.9 (8.0) days vs. 3.3 (2.4) times over 13 (8.5) days).

Table 4 details the type of ward-based nutrition support received. Exclusive or supplementary (to oral diet) EN was prescribed for 106 patients (52 (67%) in S1 and 54 (66%) in S2) for a mean of 6 (5.5) days (similar duration in both surges). More high-protein feed was used on the ward in S2 (33, 61%), compared with S1 (9, 17%) (*p* = 0.0001), whereas more high-energy feed was used in S1 (17, 33% vs. 7, 13%, *p* = 0.007). Of the 106 patients receiving EN, nasogastric feeding was ceased on the ward without prior dietetic review on 58 occasions (54%). During ward admission, 117 (73%) patients required prescribed ready-to-drink oral nutritional supplements, of which 52 (66%) were in S1 and 65 (79%) in S2. The most commonly prescribed product was a compact high-protein milkshake-style, with 32 (62%) in S1 and 40 (62%) in S2.

Of the 453 patients, 175 (39%) survived to hospital discharge, of which 128 (73%) needed community dietetic follow-up. This was consistent across both surges, with 63 (72%) in S1 vs. 65 (74%) in S2. Community referrals reasons for each surge were continued EN (15, 12%), oral nutritional supplements (70, 55%), and healthy eating advice (43, 34%).

## 4. Discussion

In this large cohort of patients with COVID-19, a standardized feeding protocol was insufficient to meet nutritional needs with individualized dietetic-led nutrition interventions required for all patients during ICU admission. Significant differences were observed in dietetic-led nutrition interventions in S1 and S2. More patients required feed adjustment for calories derived from propofol, impaired renal function, and fluid and electrolyte adjustments in S1, whereas patients more commonly needed adjustments for gastrointestinal dysfunction in S2. Over one-third of patients were malnourished on ICU admission, and patients lost an average of 7.9 kg over the ICU stay. Further weight loss was prevented over the remaining hospital stay with continued dietetic-led nutrition interventions. Only 39% of the cohort survived to hospital discharge, with 73% of these requiring further dietetic interventions in the community.

In our COVID-19 cohort, all required individualized dietetic-led nutrition interventions during ICU admission, suggesting critically ill patients with COVID-19 are nutritionally complex. This proportion is markedly higher than non-COVID patients, for whom the need for individualized dietetic-led nutrition interventions is reported to range from 50% to 70% [13,14]. Differences in dietetic-led nutrition interventions required in pandemic S1 and S2 reflect changes in medical management as knowledge of the disease evolved. Furthermore, in our institution, propofol shortages were experienced in S2, and therefore, alternate opioid sedation agents were required, obviating the need for EN manipulation to avoid overfeeding. Fluid restriction commonly employed in S1 was not used in S2, as evidence emerged that it was not beneficial [15]. Increased gastrointestinal dysfunction and EN intolerance in S2 might be attributed to two changes in clinical practice. Firstly, there was increased use of prone positioning, which can present unique feeding challenges due to large gastric residual volumes, vomiting, and aspiration of gastric contents [16]. Secondly, increased use of opioid-based sedation due to propofol shortages observed in this study may have contributed to the cycles of constipation and laxative-induced diarrhea observed [3,17].

In our patient cohort, a mean weight loss of 7.9 kg over the ICU stay was identified. Two-thirds of patients lost more than 5% of body weight, and one-third lost more than 10%, indicating severe malnutrition. Disease severity and prolonged ICU stay were likely contributing factors. Weight loss and malnutrition contribute to worse functional ability following ICU discharge in patients with COVID-19 [2,3]. A substantial proportion of patients surviving to hospital discharge requiring further dietetic-led nutrition interventions in the community was also found.

Despite all patients experiencing weight loss in ICU, no further weight loss occurred before hospital discharge, which may be attributed to ongoing dietetic-led nutrition interventions. Of the patients transferred to the ward, 90% received ward-based dietitian-prescribed individualized nutrition interventions. Continuity of nutritional interventions can be problematic during the transition from ICU to the ward [18]. Structured dietetic handover between ICU and ward dietitians to improve the transfer of nutritional history and meet nutritional needs for recovery were used [1]. Additionally, more patients received high-protein feeds on the ward during S2 as ward dietitians became more familiar with treating COVID-19 patients and as guidelines became available. A high protein diet is recommended due to the catabolism experienced during critical illness and is considered to aid recovery [1].

To our knowledge, this is the largest prospective cohort study describing dietetic-led nutritional interventions provided to patients with COVID-19 from ICU admission to ward discharge. Our study has limitations. First, it was a single-center study, which limits generalizability. Second, the observational design of the study means no assumptions about causality can be made, and therefore, we could not determine if dietetic-led nutrition interventions influenced clinical outcomes. Third, it was not possible to make comparisons to dietetic-led nutritional interventions for patients without COVID-19 at this time, as there were very few patients admitted without COVID-19. Finally, due to pandemic dietetic working conditions, data on amounts of energy and protein received could not be captured.

## 5. Conclusions

In this large, prospective cohort of patients with COVID-19, all patients required individualized dietetic-led nutrition interventions during ICU admission, as the standardized ICU nutrition protocol did not meet nutritional needs. The complexity and evolving nature of medical management necessitated multifaceted dietetic-led nutrition interventions, which differed between surges. Over one-third of patients were malnourished on ICU admission, and all patients lost weight in the ICU. Most patients were nutritionally compromised at ICU discharge and required ongoing dietitian-led individualized nutrition interventions in the ward, which prevented further weight loss over the remaining hospital stay. Based on our results, future studies are needed to determine if individualized nutrition support provision led by a dietitian increases the adequacy of nutritional delivery and improves clinical outcomes, compared with standard care.

## Figures and Tables

**Figure 1 nutrients-14-01062-f001:**
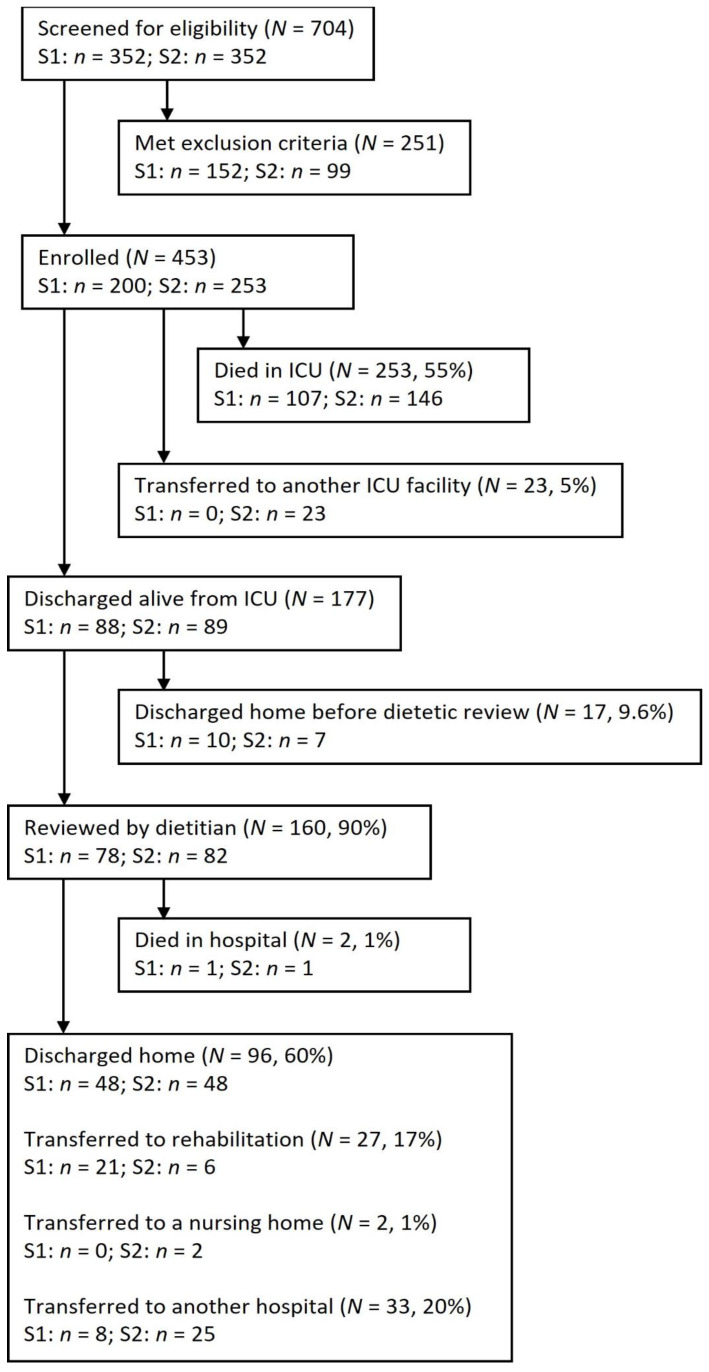
Patient flow diagram. Key—S1 = surge one; S2 = surge two.

**Table 1 nutrients-14-01062-t001:** Baseline characteristics and clinical outcomes (whole cohort and comparison between patients in S1 and S2).

Patient Characteristics	All Patients *n* = 453	S1 *n* = 200	S2 *n* = 253	*p* Value
Age (years), mean (SD)	61 (12.4)	57.9 (12.7)	62.5 (12.0)	0.001
Male gender, *n* (%)	315 (70)	135 (67)	180 (71)	0.44
Weight (kg), mean (SD)	84 (20)	86 (21)	84 (19.4)	0.32
BMI (kg/m^2^), mean (SD)	29 (6.3)	29 (6.5)	29 (6.1)	0.82
Ethnicity, *n* (%)				
White	114 (25)	46 (23)	68 (27)	0.34
Asian	96 (21)	41 (21)	55 (21)	0.85
Black	71 (16)	36 (18)	35 (14)	0.22
Other ethnic groups	48 (11)	23 (12)	25 (10)	0.57
Not stated	124 (27)	54 (27)	70 (28)	0.87
Comorbidities, *n* (%)				
Nil or 1	90 (20)	50 (25)	40 (16)	0.02
More than 2	362 (80)	150 (75)	212 (84)	
APACHE II, mean (SD)	15.1 (6.8)	15.5 (7.3)	14.8 (6.5)	0.32
ICU mortality, *n* (%)	253 (55)	107 (54)	146 (57)	0.37
ICU length of stay (days), mean (SD)	20 (18)	18.1 (14.4)	22.3 (21.2)	0.02
Total hospital stay (days), mean (SD)	35.6 (21)	33.8 (19.7)	37.4 (22.85)	0.2

Acute Physiology and Chronic Health Evaluation II score (APACHE). All values are mean (SD).

**Table 2 nutrients-14-01062-t002:** Percentage of patients requiring specific dietetic-led nutrition interventions in ICU.

Intervention	*n* = 453 *n* (%)	S1 *n* = 200 *n* (%)	S2 *n* = 253 *n* (%)	*p* Value
Feed adjustment to meet energy needs for the different metabolic phases	337 (74)	141 (71)	196 (78)	0.09
Feed adjustment to account for calories derived from propofol sedation > 15 mL/h (360 kCals/24 h)	248 (55)	144 (72)	104 (41)	0.001
Feed adjustment for gastrointestinal dysfunction	154 (34)	47 (24)	107 (42)	0.001
Transition from EN to oral diet	146 (32)	62 (31)	84 (33)	0.62
Feed adjustment due to changes in renal function, fluid status, or electrolyte balance	120 (26)	73 (37)	47 (19)	0.001
Feed adjustment to allow feed interruption for drug absorption of medication given via the enteral route	18 (4)	6 (3)	12 (5)	0.34

All values are patient numbers and (%).

**Table 3 nutrients-14-01062-t003:** Weight loss.

	All *n* = 160	S1 *n* = 78	S2 *n* = 82	*p* Value
ICU admission weight (kg)	85 (20.1)	86 (21)	84 (19.4)	0.32
ICU admission BMI (kg/m^2^)	29 (6.3)	29 (6.5)	29 (6.1)	0.82
ICU weight loss (kg)	7.9 (6.8)	7.8 (7.8)	8.1 (5.9)	0.80
ICU weight loss %	8.8 (6.9)	8.5 (7.7)	9.0 (6.3)	0.65
Percentage ICU weight loss *n* (%)				
<5%	58 (35)	30 (41)	36 (40)	0.22
5–10%	57 (34)	27 (36)	27 (30)	
>10%	53 (32)	22 (30)	26 (29)	
Total weight loss (kg) from ICU admission to hospital discharge	7.5 (6.6)	8.0 (7.4)	7.3 (6.1)	0.70

All values are mean (SD).

**Table 4 nutrients-14-01062-t004:** Nutrition support received on ward admission.

Nutrition Intervention	*n* = 160 *n* (%)	S1 *n* =78 *n* (%)	S2 *n* = 82 *n* (%)	*p* Value
Exclusive EN	36 (23)	14 (18)	22 (27)	0.17
Supplementary EN	70 (44)	38 (49)	32 (39)	0.21
Exclusive and supplementary EN combined	106 (66)	52 (67)	54 (66)	0.91
ONS	68 (43)	34 (44)	34 (42)	0.78
Texture modification	34 (21)	23 (29)	11 (13)	0.01
Diet alone	33 (21)	21 (27)	12 (14)	0.05

All values are patient numbers and (%). Enteral nutrition (EN); oral nutrition supplement drinks (ONS).

## Data Availability

Not applicable.

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
