# Peer review of "Dietetic-Led Nutrition Interventions in Patients with COVID-19 during Intensive Care and Ward-Based Rehabilitation: A Single-Center Observational Study"

_nutrients, 2022, doi:10.3390/nu14051062_

Round 1

Reviewer 1 Report

This research article is written well. No further editing is needed.

Reviewer 2 Report

This is a thoughtfully prepared research article that addresses an interesting research question.

Reviewer 3 Report

Terblanche and colleagues describe the dietitian-led nutrition interventions for patients with 12 COVID-19 during ICU and ward-based rehabilitation in a prospective single-centered study. They observed that many COVID-19 ICU patients are malnourished upon admission and that most lose significant weight during ICU stay. Their results suggest that dietetic interventions were necessary for all patients and describe how treatment trends changed over time. The authors did a great job preparing this manuscript and this reviewer has only minor edits. 

Required:

Title and throughout: Please be consistent with how the intervention is described. I would assume the only correct descriptions are dietitian-let or dietetic interventions, which would require the title to be changed. 

Throughout: Please use consistent capitalization for BMI units, m should not be capitalized. 

Throughout: Please use consistent spelling for dietitian. I am not sure if in the UK dietician is an acceptable spelling. In the US only dietitian is acceptable. Please determine which is most commonly used in the UK and spell consistently throughout manuscript. 

Lines 89-91: Please fix formatting. 

Line 78: IBW should be defined for any non-RD readers. 

Line 111: the greater than equal to symbol should be switched to a less than equal to symbol.

Line 118: Please expand this sentence. 'Informed consent was waived because...'

Line 181: Please add a space between the table and text.

Lines 211-213: Are these observations from the present study? Indicate whether this was observed in current study or in other studies. 

Table 1: I'm not sure what you mean by 'other ethnic group'. Ethnic refers to a self-identified culture, I think you mean 'other race'. But in either case, there should be a footnote describing what this category contains. 

Tables: All tables require footnotes describing the statistics used so that they stand alone. Currently lacking.

Recommended:

Figure 1. The font in this figure is different than the text. 

Lines 114-118: Did this study follow any reporting guidelines such as STROBE? If so, please add here. 

Lines 235-237: I do not perceive this as a limitation of the current study which aimed "to describe dietitian-led nutrition interventions for patients 55 with COVID-19 during ICU admission and ward-based rehabilitation in a large tertiary 56 ICU service in a single centre"

The copyright on page 1 indicates 2021. MDPI tends to update manuscript templates often, please ensure that you have the most up-to-date template. 

The impact of this work is important, but is not obvious by the discussion and conclusion as written. It would help me a lot if there was a paragraph or sentence along these lines "Based on our results, future studies are needed to..."

Reviewer 4 Report

A better study design is needed to evaluate the effectiveness of the intervention. This article compares COVID surge 1 and surge 2 as the main comparisons, rather than comparing the pre-intervention and post-intervention. So I think the experiment needs to be improved.

Author Response

A better study design is needed to evaluate the effectiveness of the intervention. This article compares COVID surge 1 and surge 2 as the main comparisons, rather than comparing the pre-intervention and post-intervention. So I think the experiment needs to be improved.

Response: Our intension was not to demonstrate effectiveness of an intervention with this study. We report observational data on a new disease for which the nutritional needs were unknown. As evidenced by other reviewer comments, we believe this study provides important insights into the nutritional needs of COIVD-19 patients and the role of the dietitian. No change to the manuscript has been made to address this point.